# Therapy of Pseudoxanthoma Elasticum: Current Knowledge and Future Perspectives

**DOI:** 10.3390/biomedicines9121895

**Published:** 2021-12-13

**Authors:** Max Jonathan Stumpf, Nadjib Schahab, Georg Nickenig, Dirk Skowasch, Christian Alexander Schaefer

**Affiliations:** Medical Department II, Cardiology, Pneumology, Angiology, University Hospital Bonn, 53127 Bonn, Germany; Nadjib.Schahab@ukbonn.de (N.S.); georg.nickenig@ukbonn.de (G.N.); Dirk.Skowasch@ukbonn.de (D.S.); Christian.Schaefer@ukbonn.de (C.A.S.)

**Keywords:** pseudoxanthoma elasticum, PXE, therapy, skin, choroidal neovascularization, atherosclerosis, pyrophosphate, gene therapy

## Abstract

Pseudoxanthoma elasticum (PXE) is a rare, genetic, metabolic disease with an estimated prevalence of between 1 per 25,000 and 56,000. Its main hallmarks are characteristic skin lesions, development of choroidal neovascularization, and early-onset arterial calcification accompanied by a severe reduction in quality-of-life. Underlying the pathology are recessively transmitted pathogenic variants of the *ABCC6* gene, which results in a deficiency of ABCC6 protein. This results in reduced levels of peripheral pyrophosphate, a strong inhibitor of peripheral calcification, but also dysregulation of blood lipids. Although various treatment options have emerged during the last 20 years, many are either already outdated or not yet ready to be applied generally. Clinical physicians often are left stranded while patients suffer from the consequences of outdated therapies, or feel unrecognized by their attending doctors who may feel uncertain about using new therapeutic approaches or not even know about them. In this review, we summarize the broad spectrum of treatment options for PXE, focusing on currently available clinical options, the latest research and development, and future perspectives.

## 1. Introduction

Pseudoxanthoma elasticum (PXE, OMIM 264800) is a rare, genetic disease with autosomal recessive inheritance. Its prevalence is estimated to be between 1 in 25,000 and 56,000 of the general population [1,2]. The gene/protein affected by the mutation, ATP-binding cassette subfamily C member 6 (*ABCC6*/ABCC6), is an adenosine triphosphate (ATP)-dependent transmembrane transporter, which is mainly expressed in the liver and kidneys, but also, to a minor degree, in peripheral tissues. ABCC6 is encoded by the *ABCC6* gene located on chromosome 16, location p13.1 [3,4]. Although its specific substrate is still unknown, ABCC6 is involved in the homoeostasis of serum pyrophosphate (PPi), a main inhibitor of ectopic calcification. Dysfunctional or missing ABCC6 results in low serum PPi and, thus, ectopic calcification, especially of soft connective tissue. 

The clinical appearance of PXE mainly consists of several hallmarks: formation of characteristic alterations to the skin, varying from yellowish coalescing papules to lax and redundant skin, mainly around the neck and flexural areas of the main joints [5,6]; in addition, the development of angioid streaks and neovascularization in the retina followed by a retinal hemorrhage and scarring, with gradual loss of central and night vision and, ultimately, blindness, are symptomatic of PXE [7,8]; also, early onset atherosclerosis with increased vascular occlusion and a significant burden of intermittent claudication can be signs of PXE [9,10,11,12]. Microvascular alterations [13] and impaired pulmonary carbon monoxide diffusion capacity are further phenotypic aspects [14]. In particular, the loss of vision and vascular disease result in a progressively impaired quality-of-life and reduced participation in social and occupational activities [15]. 

Some previous attempts at reviewing the current knowledge of PXE have been made. Amongst these, two publications should be highlighted: Dominique Germain published a review in 2017 that covered the current information to the full extent [16]. Recently, Shimada et al. contributed an impressive review of new therapeutic solutions to treat PXE [17]. However, from a clinical point of view multiple questions remained unanswered: how to treat PXE patients in the daily clinical routine? Are interventional therapies for atherosclerotic lesions in cardiac and peripheral arteries safe and effective? How should we perform oral anticoagulation therapy in patients with PXE? These and other questions have never been fully addressed. Therefore, the main intention of this review is to focus on current treatment possibilities regarding the skin changes, choroidal neovascularization, and atherosclerosis. However, as new therapeutic advances, such as pyrophosphate substitution or bisphosphonate administration, will probably become an integral part of PXE treatment, they also will be addressed in this review. As this review intends to comprehensively overview possible therapeutic options that clinical physicians are frequently asked about, other therapeutic approaches with—from today’s point view—little or no chance of ever being used as a therapeutic agent for PXE, will also be summarized.

## 2. Literature Research

A comprehensive search of the Medline/PubMed database was conducted. The last query was on 1 November 2021. Findings were limited to English and German articles. Articles published before 1980 were excluded, unless no newer information has been published on a topic, or their inclusion was necessary for understanding. Due to the wide spectrum of this article, several different search terms were used in the following schema: ((Pseudoxanthoma elasticum) OR (PXE)) AND (XXX). Moreover, references of the articles cited were searched for, to find further publications not detected by the literature search. All of the articles identified were evaluated by their title and/or abstract and included if considered appropriate.

## 3. Therapeutic Approaches

PXE is a hereditary disease and, so far, there is no cure for it. Until recently, the therapeutic approaches used were mainly symptom-oriented. New therapeutic advances, however, focus on systemic therapy, using bisphosphonates or pyrophosphate to reconstitute the homeostasis of calcification. This chapter summarizes the current and recent therapeutic possibilities for the affected organ systems, and surveys new therapeutic advances.

### 3.1. Therapy for Skin Lesions

Commonly, skin lesions are the first phenotypic manifestation of PXE [18]. They initially consist of small, localized, and yellowish- or skin-colored papules. With adolescence, the papules progressively coalesce into larger, lax and redundant plaques of skin, mainly localized around the neck and flexural areas of the main joints, but also into oral, vaginal, and rectal mucosae [16,19]. Although the skin alterations are mostly asymptomatic, there are a few cases of perforating PXE, characterized by chronic or recurrent ulceration of skin lesions [20,21]. Beyond that, alterations of the skin may individually be experienced as disfiguring, especially by female patients. 

Data on the treatment of skin lesions in patients with PXE are scarce. Three different approaches have been reported: cosmetic surgery, injection of collagen, and resurfacing of the skin by using a fractional CO_2_ laser.

#### 3.1.1. Cosmetic Surgery

There are a few case reports and case series on cosmetic surgery in patients with PXE [22,23,24,25,26]. The surgery performed is usually the standard procedure with a subcutaneous rhytidectomy and neck skin lifting [19], with satisfactory primary results. However, poor wound healing, friable skin, and keloid formation are recurrent problems that should not be underestimated.

#### 3.1.2. Injectable Collagen

Attempts have been made at treating horizontal chin creases with injectable collagen, and in two cases, the treatment showed temporary improvement for up to a few months [27]. 

#### 3.1.3. Fractional CO_2_ Laser

There is one reported case of skin resurfacing by using a fractional CO_2_ laser [28]. Herein, multiple applications were necessary with only moderate improvement. Side effects were temporal hyperpigmentation and inflammation of the skin, as well as a recurrence of a local herpes simplex.

#### 3.1.4. Conclusions

Given that skin lesions in PXE were first described in 1881 by Rigal, and then in 1896 by Darier [16,29], the small number of reports on therapy for this condition is surprising. In addition, there are only a few cases with satisfactory or moderate primary results, this is suggestive of a positive selection bias. Since a prospective study with structured follow-up has never been conducted for any kind of treatment of skin lesions in PXE, these therapies should be considered with caution and only used after carefully weighing the operative risks and individual suffering. 

### 3.2. Therapy for Choroidal Neovascularization

Impending ocular manifestations and a progressive loss of vision is often a stressful and frightening situation for patients with PXE. Underlying this pathology is the progressive calcification and friability of Bruch’s membrane (BM) [7]. Anatomically, BM is located between the retinal pigment epithelium (RPE) and the choriocapillaris. RPE is a monolayer of cells directly adjacent to the photoreceptors, its function is to reduce the scattering of light and mediate metabolism and nutrient exchange for the cones and rods via diffusion through the BM. The choriocapillaris is heavily capillarized and provides nourishment to the outer retinal layers and the BM. The BM itself serves as a smooth and wrinkle-free surface for the RPE and retina, and as a physiological barrier against capillaries growing into the retina [30]. Progressive calcification leads to breaks in the BM, which are visible as angioid streaks in fundoscopy [31,32]. With a progressive loss of function of the BM, choroidal neovascularization (CNV) through the BM, especially in the macular region, becomes the main ocular manifestation of PXE. Untreated, CNV results in local subretinal bleeding and exudation, leading to fibrovascular scarring and a progressive loss of vision [7,31]. As there are no prophylaxes for CNV to date, current and recent therapeutic approaches have targeted the neovascular membranes to prevent subretinal bleeding.

There are five different approaches to treat CNV in PXE, of which the majority are considered outdated. These approaches include laser photocoagulation, transpupillary thermotherapy, photodynamic therapy, surgery, and inhibition of vascular endothelial growth factor (anti-VEGF) treatment. 

#### 3.2.1. Laser Photocoagulation

Laser photocoagulation (LPC), using argon, krypton, or dye laser technology, was the first successful attempt at treating choroidal neovascularization, secondary to age-related macular degeneration (AMD). LPC in AMD showed early beneficial effects that persisted for at least five years in the follow-up [33]. Although LPC has become less important for treating AMD, in favor of anti-VEGF-treatment, it still is frequently applied [34,35]. Therefore, the idea of using LPC in patients with PXE was obvious. However, there has never been a prospective or retrospective analysis regarding the usage of laser photocoagulation in PXE only. Presented in 1981 as a case series focused on patients with angioid streaks in general, Singerman et al. [36] reported eight cases of patients with angioid streaks that were treated with LPC, of which six patient presented with skin changes consistent with PXE. The average follow-up was three years, and the authors were able to successfully obliterate most neovascular membranes. However, the appearance of new neovascularizations was common. It was concluded that the application of LPC in carefully selected cases, without affection of the fovea, should be considered; follow-up examinations at close intervals were recommended due to the reoccurrence of the neovascular membranes. Brancato et al. (1987) presented a series of 31 patients with angioid streaks, of which 20 were affected with PXE [37]. The mean follow-up was 15.6 months and, in total, 60 eyes were examined. Thirteen out of thirty-nine eyes with subretinal neovascularizations were treated with LPC. Ten of these eyes presented with one or more recurrences within three months; five eyes were retreated. The authors reported that treated patients had a better final visual acuity compared to untreated patients. Deductively, the authors deemed that photocoagulation was worth performing in selected patients, with close follow-up examinations. Lim et al. reported a total number 20 patients and 24 eyes with active choroidal neovascularization that underwent LPC [38]. Fourteen of these patients suffered from PXE. Within a mean post-treatment follow-up period of 3.5 years, CNV persisted in five eyes after treatment, and CNV reoccurred in an additional nine eyes. The authors concluded that LPC should be considered in well-demarcated cases of CNV not involving the fovea. The largest evaluation yet of LPC in patients with angioid streaks was published by Pece et al. in 1997 (abstract only) [39], who surveyed 66 eyes from 52 patients that were treated with laser LPC due to CNV secondary to angioid streaks. Although the final visual acuity in the treated eyes was higher compared to untreated eyes, a recurrence of CNV was observed in 77% of the treated eyes.

In summary, LPC in CNV secondary to angioid streaks currently is considered an outdated treatment, especially because its non-applicability in subfoveal CNV substantially limits the value of LPC for angioid streaks and CNV associated with PXE.

#### 3.2.2. Transpupillary Thermotherapy

Transpupillary thermotherapy (TTT) also is a laser-based technology. In contrast to LPC, TTT uses an infrared laser and has a lower target temperature [40]. Since 1999, it has been applied on patients with CNV due to AMD; the results, however, have been discouraging [40]. Data on the treatment of CNV due to angioid streaks with TTT are scarce. Aras et al. (2004) reported two cases; both involved patients that showed disease progression despite treatment [41]. In 2006, Özdek et al. presented a series of five cases [42]. They observed an initial decrease in CNV activity after treatment, but the patients needed retreatment and there were unfavorable results thereafter. 

#### 3.2.3. Photodynamic Therapy

Photodynamic therapy (PDT) is based on intravenous injection of a photosensitive dye (Verteporfin), which accumulates in neovascular endothelial cells. After selective activation of the dye via laser light, the emitted light induces local inflammation and vascular occlusion [43]. The main advantage of this treatment, over other laser-based procedures, is the possibility to selectively target neovascular capillaries. 

There have been various retrospective and prospective case series investigating PDT for the treatment of CNV due to angioid streaks [44,45,46,47,48,49,50,51,52,53,54,55]. The surveyed literature comprised a total number of 157 patients and 163 eyes treated with PDT. Of those, if reported, 51 patients were diagnosed with PXE. Presumably due to small sample sizes, the reports were somewhat inconsistent. The first report of PDT treatment in patients with CNV due to angioid streaks was published by Karacorlu et al. in 2002 [44]. The authors surveyed eight patients with angioid streaks associated with PXE. Herein, a cessation or decrease of fluorescein leakage was observed; PDT was considered an adequate treatment for CNV due to angioid streaks. In contrast, Shaikh et al. (2003) reported no relevant alteration of the course of the disease, despite multiple PDT treatments. The largest evaluation of PDT for the treatment of CNV associated with angioid streaks was conducted by Menchini et al. in 2004, who presented a series of 40 cases [45]. They observed a significant loss of vision during the first six months after the initial treatment.

In summary, most of the previous reports emphasized possible benefits of PDT compared to the natural course of CNV. However, the majority of authors concluded that PDT is not an effective treatment for CNV associated with angioid streaks. Especially in patients with subfoveal neovascularization, there was a high risk of disease progression despite treatment.

#### 3.2.4. Surgery

Macular translocation surgery (MTS) is an operation that attempts to move the central sensory retina, including the fovea, from an area of diseased RPE or BM to a less affected area. A combination with other approaches, such as laser photocoagulation of the remaining choroidal neovascular membranes, is common. Initially it has been performed on patients with AMD [56]. However, it also has been used to treat non-AMD subfoveal CNV [57]. A literature search revealed a total of six patients with CNV due to angioid streaks that have been treated with MTS [57,58,59]; a diagnosis of PXE was reported in one case [59]. Fujii et al. reported four cases of CNV due to angioid streaks that were treated by MTS [58]. Of those, two patients presented without a recurrence of CNV during the follow-up. However, one patient suffered from multiple recurrences, and another sustained an intraoperative submacular hemorrhage and postoperative retinal detachment. Roth et al. reported a combination of MTS and laser photocoagulation in one patient with PXE, who successfully underwent surgery and presented as stable during the follow-up [59]. Ehlers et al. published one case of CNV due to angioid streaks [57]. Herein, the patient developed further loss of vision after MTS. 

Surgical removal of choroidal neovascular membranes (SRCNV) describes the operational resection of CNV via retinotomy. Eleven patients who underwent this procedure were identified in a literature search [60,61,62,63]. Thomas et al. (1993) reported four cases of CNV associated with angioid streaks [60]. Loss of vision continued after the operation. However, the included patients were presented with an already highly impaired vision at the baseline. Recurrence of CNV was observed in one of these patients. Adelberg et al. published five cases with angioid streaks due to PXE [61]. Herein, one patient was presented with an improved postoperative vision, another one sustained further loss of vision, and no change occurred in the remaining cases. Complications from the surgery were intraoperative retinal bleeding and peripheral retinal detachment. Eckstein et al. added a report of one patient with a worsening of visual acuity after treatment and a recurrence of CNV after two months [62].

Parolini et al. (2016) reported one case of CNV due to angioid streaks that was successfully treated by using an autologous RPE and choroidal patch [64]. After multiple, but unsatisfactory, intravitreal anti-VEGF injections, surgical treatment achieved an improvement of vision and restored the patient’s ability to read.

In summary, surgical approaches can achieve short-term stabilization or even improvement of visual acuity. With the progressive nature of CNV associated with angioid streaks, recurrences are frequently observed, and complications such as hemorrhage or retinal detachment limit the value of these surgical approaches. Furthermore, the transferability of patients with PXE is questionable, as most reported cases presented with angioid streaks without a proven diagnosis of PXE.

#### 3.2.5. Anti-VEGF Treatment

Vascular endothelial growth factor (VEGF) is a key player in the coordination of neurovascular hemostasis, particularly in vasculogenesis [65]. Intravitreal injection of vascular endothelial growth factor inhibitors (anti-VEGF) for the treatment of CNV was first applied to patients with AMD in 2005 [66]. Since then, it has been proven to be effective in reducing the CNV disease, consecutive bleeding, and progressive loss of vision. Previously common therapies, such as laser photocoagulation or PDT, have been widely replaced by anti-VEGF treatment [66,67]. The first case of intravitreal anti-VEGF treatment in a patient with CNV due to angioid streaks was reported by Teixeira et al. in 2006 [68]. Herein, the authors described a case of acute vision loss in the right eye due to subfoveal CNV with a subretinal hemorrhage and subsensory fluid. After two intravitreal injections (at the baseline and after seven weeks) with bevacizumab, visual acuity (VA) was largely restored. As a second case, CNV secondary to angioid streaks was reported by Lommatzsch et al. in 2007 [69]. Here, the authors presented a patient with PXE and consecutive CNV that was successfully treated with a three-fold intravitreal injection of bevacizumab, after multiple unsuccessful treatments with PDT. Since then, intravitreal anti-VEGF treatment has been the subject of multiple case reports, case series, and clinical trials and has emerged as the first-choice treatment for CNV secondary to angioid streaks and the ocular manifestations of PXE [68,69,70,71,72,73,74,75,76,77,78,79,80,81,82,83,84,85,86,87,88] (Table 1).

Three anti-VEGF agents are currently in use: bevacizumab (off-label), ranibizumab, and aflibercept [89]. Amongst these, the most data are available on bevacizumab. The application of these drugs varies between the substances. Bevacizumab is usually applied following a pro re nata (PRN) regimen. Here, intravitreal injection is performed according to disease activity, and assessed via best-corrected visual acuity and optical coherence tomography (OCT) [90]. Ranibizumab is usually injected with a loading dose (e.g., four injections in monthly intervals) followed by a PRN regimen. Alternatively, Ladas et al. implemented an “inject and extend” regimen [78], adapted from Spaide and his colleagues [91]. Ladas and his associates administered four injections every four weeks, and scheduled the next follow-up depending on disease activity. If the CNV was still active after the fourth injection, the next appointment would be after four weeks; if no CNV activity was apparent the next appointment would be in six weeks. This approach allows for individually adjusted treatment according to CNV activity. In contrast to bevacizumab and ranibizumab, which mainly target VEGF-A, aflibercept inhibits VEGF-A, VEGF-B, and placental growth factor (another VEGF family member) [65] and, therefore, is used in patients with persistent or recurrent CNV despite treatment with bevacizumab or ranibizumab [88].

Adverse events amongst these studies were rare: two patients suffered lens injury and one patient sustained an ischemic stroke. Eye pain, acute decrease of visual acuity, amaurosis, and infections were also reported in isolated cases. Recurrence of CNV was frequently observed; therefore, patients should be monitored in monthly intervals after the first injection. The extension of intervals is possible depending on the clinical course and the results of OCT. In general, the intravitreal injection of anti-VEGF agents is associated with endophthalmitis, intraocular inflammation, rhegmatogenous retinal detachment, intraocular pressure elevation, and ocular hemorrhage [92]. All intravitreally administered anti-VEGF agents can be detected in systemic circulation [93]; therefore, systemic adverse events due to a peripheral blockage of VEGF are possible. In particular, thromboembolic events have been associated with all anti-VEGF agents [92,93]. Other systemic adverse events of anti-VEGF treatment are non-ocular hemorrhage, acute elevation of blood pressure, stroke, myocardial infarction, arterial aneurysm, and death [92]. Another complication of intravitreal administration of anti-VEGF agents is the development of geographic atrophy (GA) of the RPE [80]. This risk seems to be associated with the frequency and number of treatments [94]. Although a combination of PDT and intravitreal anti-VEGF treatment did not show an additive effect on CNV [95], larger studies investigating combination treatments to reduce the number and frequency of injections should be considered in the context of GA.

#### 3.2.6. Conclusions

Due to the effectiveness and safety of the intravitreal administration of anti-VEGF agents, anti-VEGF treatment has rapidly become state-of-the-art, whereas other therapeutic approaches have largely been abandoned. One of the main advantages of anti-VEGF treatment, especially over the laser-based procedures, is its applicability in subfoveal CNV. Anti-VEGF treatment should be initiated at the first suspicion of subretinal fluid or neovascularization to achieve the best possible results. Patients should be educated in using self-tests, such as the Amsler grid test, to notice visual impairment at the earliest possible moment. Surgical therapy may be a rescue option in patients with refractory subfoveal CNV, despite anti-VEGF treatment, and an imminent risk of blindness.

### 3.3. Therapy for Arterial Calcification

Arterial calcification is one of the main phenotypic hallmarks of PXE [16], affecting most parts of the arterial tree, including capillaries, while the aorta is relatively unaffected [9,10,11,12,13,96,97,98]. The morphology of arterial calcification in PXE is mixed. Medial calcification, comparable to patients with diabetes or chronic kidney disease [96], and intimal calcification mixed with the stenosis and occlusion of peripheral arteries [10] often exist simultaneously. “Regular” atherosclerosis is induced by an accumulation of low-density lipoprotein (LDL) in the intima, resulting in monocyte-induced inflammation [99] and, subsequently, calcification. In PXE, atherogenesis is not yet fully understood. Although positron-emission tomography scans did not assess vascular inflammation in PXE patients [100], the presence of capillary alterations may indicate a role for the hypoxia-induced formation of reactive oxygen species as a second messenger in the pathophysiology of PXE [13]. A dysregulation of blood lipids also is part of the PXE phenotype [12,101,102]. It appears that atherogenesis in PXE is a multifactorial process, with similarities to “regular” but accelerated atherogenesis that is accompanied by rapid calcification. Calcification and fragmentation of internal and external elastic laminae [96] further promote atherogenesis due to disintegration and vulnerability of the intima [103].

Possible consequences of arterial calcification are peripheral artery disease (PAD) and intermittent claudication, coronary artery disease (CAD) and a myocardial infarction, and an ischemic stroke. 

PAD in PXE has been well characterized [9,11,12]. Although atherosclerotic lesions in PXE can potentially affect most vascular territories, arteries of the lower extremities are usually the most severely altered. A high atherosclerotic burden and high prevalence of arterial occlusion lead to intermittent claudication, reduced social participation, and an overall lower quality-of-life in these patients [10,104].

The significance of CAD in PXE is less clear. Although this seems to be an obvious comorbidity, the association between PXE and CAD originates mainly from case reports [105,106,107,108,109,110,111,112,113,114,115,116,117]. Furthermore, one of the pathogenic variants that frequently causes PXE, the nonsense mutation c.3421C>T (p.Arg1141*) on the *ABCC6* gene, has been connected to an increased risk of premature CAD in heterozygous carriers [118,119]. Clinical studies and autopsy cases have, however, failed to prove an elevated prevalence of CAD in PXE patients [96,97,120,121]. Nonetheless, Kranenburg et al. detected arterial calcification of coronary arteries, as assessed by a CT scan, in 54% of PXE patients [11]. Therefore, CAD is surely part of the PXE phenotype. Cardiac function may also be impaired, as the subendocardial tissue in PXE is frequently described as thickened, with high numbers of degenerated and fragmented elastic fibers [96,97,104,121]. Despite these histological characteristics, reports of an impaired left-ventricular function are inconsistent. Campens et al. reported an impaired myocardial relaxation in 32 patients with PXE [122]. On the opposite side, Bière et al. detected no cardiac dysfunction in a 75-patient cohort of PXE [123]. The latter study was supported by Prunier and his associates, who identified no frequent cardiac complications in a cohort of 67 patients with PXE [120]. In their publication, the authors also reported cardiac findings from *ABCC_-/-_* mice, which developed cardiac hypertrophy without contractile dysfunction. This was interpreted as a hint that PXE patients might be at risk of developing cardiomyopathy as they become older [120]. In summary, symptomatic CAD and cardiac dysfunction do occur within PXE, although elevated rates of ischemic cardiac events have not yet been demonstrated.

#### 3.3.1. General Recommendations and Walking Exercise

In general, the lifestyle recommendation for patients with chronic coronary syndrome can also be applied to patients with PXE [124]. This includes smoking cessation, a healthy, Mediterranean diet, physical activity, preferably on a regular basis, weight control (body mass index < 25 kg/m²), taking medications as prescribed, and the treatment of comorbidities, such as arterial hypertension (AHT) and diabetes. The treatment of comorbidities in PXE patients focuses, in particular, on diseases with an inherent cardiovascular risk. This is fundamental, since these comorbidities, especially AHT, seem to be associated with accelerated arterial calcification in PXE [125]. Blood pressure should be treated to achieve a systolic target of 130/80 mmHg [124].

Regarding PAD, and intermittent claudication in particular, physical activity should include exercise (walking exercise) [126,127]. Three possible approaches are commonly defined: supervised exercise therapy (SET), home-based exercise therapy (HBET), and walking advice (WA). SET describes walking exercise with face-to-face instruction on a regular basis. HBET is defined as structured walking advice with an observational component, such as logbooks or a pedometer. A “go-home-and-walk” advice without instruction and observation is considered walking advice [128]. Of those, SET provides the best results, with an average improvement of walking distance between 120 and 210 m, whereas data on HBET and, especially, WA are less favorable [128]. Walking exercise in patients with PXE is crucial, as PAD usually occurs prematurely. This results in mostly well-compensated PAD with efficient collateralization [10,103]. Walking exercise, especially SET, promotes muscle capillarization, induces angiogenesis, and, subsequently, collateralization [127].

#### 3.3.2. Medical Treatment of Peripheral and Coronary Artery Disease

At present, medical treatment of arterial calcification generally follows the current guidelines [124,127].

Although new therapeutic approaches, such as bisphosphonate or pyrophosphate administration, may alter medical treatment in the future, lipid-lowering drugs are currently the state-of-the-art therapy for PXE. This is not only according to current guidelines [126], but is also rooted in pathology of PXE: an in silico analysis by Hosen et al. (2014) identified various possible substrates of the ABCC6 transporter, of which lipids and bile acids were found to be the most likely [129]; Kuzaj and his associates (2014) reported increased cholesterol biosynthesis and elevated levels of proprotein convertase subtilisin/kexin type 9 (PCSK9) in PXE fibroblasts [130]; recently, Ibold and his colleagues provided evidence of upregulated cholesterol biosynthesis and elevated levels of PCSK9 in *ABCC_−/−_* mice [101]. Therefore, it can be assumed that cholesterol homeostasis is at least partly involved in arterial calcification in PXE [12,101,102].

In PXE, administration of atorvastatin is favored over other cholesterol-lowering drugs because Guo et al. (2013) reported prevention, but not reversion, of ectopic calcification in *ABCC_−/−_* mice after the administration of atorvastatin [131]. As PXE patients are at a high risk of developing arterial calcification, a target value of LDL < 1. mmol/L (55 mg/dL) should be used, as is applied to patients with diabetes and a very high cardiovascular risk [132]. To achieve appropriate LDL levels, a combination of ezetimibe, bempedoic acid, and/or PCSK9 inhibitors should be considered.

#### 3.3.3. Antiplatelet (AP) and Anticoagulant (AC) Drugs in PXE

The treatment of PXE patients with anticoagulation therapy needs to be intensively discussed between physicians and patients. The administration of antiplatelet (AP) drugs (e.g., aspirin) is especially connected with fear of retinal bleeding and a progressive loss of vision. However, to the best of our knowledge, there are no convincing reports of retinal bleeding in PXE that are associated with AP or AC therapy. With regard to AMD and the risk of subretinal bleeding associated with AP or AC, reports are inconsistent. Kuhli-Hattenbach et al. [133] reported an increased risk of a severe subretinal hemorrhage in AMD patients receiving AP or AC in an observational cohort of 71 AMD patients with and without AP/AC. Patients with comorbid AHT were more severely affected. Ying et al. [134] published a clinical trial that included 1185 patients with and without AP or AC therapy. Herein, there was no increased risk associated with anticoagulation therapy. Notably, in patients administered with AP or AC and comorbid AHT, they were found to have an increased rate of subretinal bleeding. Buitendijk and his associates [135] retrospectively surveyed 330 patients with and without AP or AC, and detected no association of AP or AC with either visual decline or the occurrence of hemorrhages. A large-scale clinical trial, investigating the effect of aspirin on AMD incidence, progression, and subretinal hemorrhages, is currently underway and hopefully will provide further evidence in this matter [136,137].

Additional attention should be paid to coumarin derivates, such as warfarin or phenprocoumon, as vitamin K antagonists, associated with accelerated medial arterial calcification [138]. The administration of warfarin in *ABCC_-/-_* mice resulted in highly increased calcification of various tissues [139]. In addition, the inhibition of vitamin K 2,3-epoxide reductase complex subunit 1 (VKORC1)—which is the main target of coumarin derivates—has been connected to severe medial arterial calcification in rats [140]. Therefore, coumarin derivates should only be used with extreme caution in patients with PXE, if at all.

In conclusion, if AP/AC administration is indicated, the prescribing physicians should be aware of possible subretinal bleeding in patients with PXE. Preceding treatment of AHT to achieve the target values seems to be crucial to avoid further harm. The patient should be aware of a possible risk, and ophthalmologic monitoring should be carried out at shorter intervals. Coumarin derivates are to be avoided, if possible.

#### 3.3.4. Interventional and Surgical Therapy for PAD in PXE

As mentioned above, PAD in PXE patients is often well-compensated with efficient collateral circulation [10,103]. Therefore, interventional therapy is rarely necessary. Accordingly, reports on interventional therapy for PAD in PXE are rare. A literature search revealed only seven cases published to date [106,141,142,143]. Carter et al. reported one case, with distal occlusion of the left femoral artery, that was successfully treated by femoropopliteal bypass grafting. Patent bypass flow after 12 months was documented [141]. Slade and his associates presented a case with severe intermittent claudication of the right leg. Percutaneous transluminal angioplasty (PTA) of a high-grade stenosis of the femoral artery was performed with satisfying primary results; however, no follow-up has been reported [106]. Rühlmann et al. published one case with arterial occlusion of both femoral arteries, of which the right side had already been successfully treated by femoropopliteal bypass grafting. Surgical endarterectomy of the left femoral artery was performed with satisfying primary results. After four months, the patient suffered from total re-occlusion and was treated conservatively thereafter [142]. Ammi et al. presented a series of four cases, each treated by PTA initially. All patients sustained multiple short-term re-occlusions or -stenoses. Following this, two patients were treated conservatively, because they had potent collateral circulation, and two patients were treated by femoropopliteal bypass grafting. The authors concluded that angioplasty and the stenting of femoral arteries in PXE patients should not be recommended [143].

In conclusion, the stenosis and occlusion of femoral arteries can often be treated conservatively by walking exercise and medical treatment. In cases of severe intermittent claudication, interventional revascularization or femoropopliteal bypass grafting are an option. Vascular specialists should feel encouraged to publish further cases of interventional or surgical revascularizations in patients with PXE, as current data does not allow for a sufficient assessment of symptomatic vascular occlusion in PXE.

#### 3.3.5. Interventional and Surgical Therapy for CAD in PXE

Although symptomatic CAD is not a hallmark of PXE [120], calcifications of coronary arteries are common in patients with PXE [11,103]. Accordingly, reports on coronary revascularization, by either percutaneous coronary intervention (PCI) or coronary artery bypass grafting (CABG), are rare (Table 2). There have also been no case series or studies of any kind reporting on more than one patient. A literature search resulted in a total number of 13 reported cases [105,106,107,108,109,110,111,112,113,114,115,116,117], including three patients who were treated conservatively. Five of these cases included patients below 35 years. This further lessens the significance of the results, since severe PAD/CAD is uncommon in PXE patients of that age. Therefore, with the current knowledge of the disease available, we suspect that these patients may have suffered from PXE/GACI-overlap syndrome, which is due to mutations in both the *ABCC6* and *ENPP1* genes [144].

Conservatively treated patients had a mean age of 26 years and presented pronounced collateralization [106,107,111]. Symptoms were controllable via medical treatment. Only Kévorkian et al. reported a two-year follow-up at which the patients presented as symptom-free [107].

CABG was reported in five cases with a mean age of 44.6 years [105,109,112,113,116]. Accurate preoperative evaluation of arterial grafts is necessary because internal mammary arteries may also be affected by arterial calcification [112,113]. Song et al. reported the use of the left internal mammary artery and both radial arteries for the CABG procedure [116]. After one year, both radial grafts were occluded or had high-grade stenosis; revascularization via PCI was necessary. Therefore, radial artery grafts should be used with caution, especially as arteries of the upper extremities also are frequently affected from arterial calcification in PXE [103].

PCI was performed in five cases (mean age: 48.2 years) [108,110,114,115,117]. No peri-interventional events were reported. A twelve-month follow-up has only been reported in one case without any signs of re-occlusion [110]. Notably, two authors reported difficulties during stent implantation because sequential high-pressure post-dilations using non-compliant balloons with up to 30 atmospheres of pressure were necessary to optimize stent expansion [114,117]. Intravascular ultrasound (IVUS) did not reveal an acoustic shadow as a correlate of calcification; therefore, the lesions were assumed to consist of dense fibrotic tissue [114]. Consistent with this, Miwa et al. reported a five-layer appearance of coronary arteries, and an additional echo signal reflected from between the internal and external elastic laminae, without acoustic shadowing [145]. This is in line with Mendelsohn et al. who described fragmentation of the internal and external elastic laminae in peripheral muscular arteries. Furthermore, the authors mentioned fibrous thickening and occlusion of radial arteries [96].

#### 3.3.6. Conclusions

At present, therapy for arterial calcification in PXE follows current guidelines for PAD, CAD, and ischemic strokes. This includes lipid-lowering therapy with statins, treatment of comorbidities, and exercise training, as well as, at least, yearly examinations by specialized angiologists and cardiologists. Anticoagulation and antiplatelet therapy should also be prescribed, if necessary, but only after treatment of comorbid AHT; great caution is recommended due to the risk of retinal bleeding. Data on the revascularization of peripheral and coronary arteries are scarce. More studies are needed to gain further insight into the specifics of peripheral and coronary artery interventions in patients with PXE.

### 3.4. Supplementary and Dietary Approaches

Various supplementary and dietary approaches have been discussed during the last two decades as a possible treatment for PXE (Table 3). The idea of reduced levels of an unknown circulating anti-mineralization factor originated from parabiosis experiments with *ABCC6_−/−_* and wild type mice [146,147], implicating PXE as a metabolic disease. As in vertebrates, extracellular bodily fluids are supersaturated with calcium and phosphate, with a resulting tendency towards the spontaneous formation of calcium phosphate, meaning potent inhibitors of ectopic mineralization are crucial for survival [148]. Therefore, researchers have long looked for an impaired inhibitor of soft-tissue calcification that is easy to supply.

#### 3.4.1. Calcium and Phosphate

A connection between calcium intake and ectopic calcification was already proposed in 1984 by Renie et al. [149], who correlated the presumed calcium intake during childhood, adolescence, and adulthood with severity of the disease in adulthood. They found that a high intake of calcium during childhood and adolescence positively correlated with the overall severity of the disease in adulthood. Despite its strong limitations (e.g., estimated calcium intake) and its irreproducibility, the idea of calcium intake determining disease severity has persisted, at least until the development of the murine model of PXE [150]. In this context, Gorgels et al. found no effect of calcium intake on vascular calcification in *ABCC6_−/−_* mice [151]. Scherer et al. investigated a possible role of oral phosphate binders in the treatment of PXE by administering aluminum hydroxide, a phosphate binder, to six patients with PXE. Although skin changes seemed to show clinical improvement in three out of six patients, no improvement was observed with regard to fundoscopic changes [152]. In 2011, Yoo et al. conducted a randomized, double-blind, placebo-controlled study including 40 patients with PXE [153]. Herein, the administration of the phosphate binder, sevelamer hydrochloride, resulted in a reduction of calcification levels. However, no significant differences occurred compared to the placebo. The authors suspected a possible confounding effect of magnesium intake. 

#### 3.4.2. Magnesium

The protective role of magnesium (Mg^2+^) in cardiovascular mortality and vascular calcification has been repeatedly reported [154]. Its protective effect seems to be mediated via passive buffering of monophosphate and reduced hydroxyapatite formation, leading to actively altered osteogenic expression in vascular smooth muscle cells [154]. Moreover, Mg^2+^ deficiency is related to various cardiovascular comorbidities that are also reported in patients with PXE, such as arterial hypertension, atherosclerosis, endothelial dysfunction, and arterial calcification [155]. On the other hand, Mg^2+^ protects elastic fibers from calcium deposition and, therefore, helps to preserve the physiological function of soft connective tissue [155]. Especially in patients with chronic kidney disease (CKD), administration of Mg^2+^ shows protective effects against vascular alterations [154], which is important since CKD and PXE share some vascular phenotypes [156]. A possible role for magnesium in PXE therapy has been discussed since LaRusso et al. (2008) reported an increased mineralization of soft connective tissue in *ABCC6^−/−^* mice fed with an experimental diet containing decreased amounts of Mg^2+^ [157]. This effect was confirmed by the authors in 2012 [158]. Moreover, an Mg^2+^-enriched diet has repeatedly been reported to prevent connective tissue mineralization in *ABCC6_−/−_* mice [151,159,160]. The administration of Mg^2+^ was accompanied by a reduction of carotid intima-media thickness in *ABCC6_−/−_* mice [161,162]. Supported by these promising results, Rose et al. conducted a randomized, double-blind, placebo-controlled prospective trial to evaluate the effect of oral magnesium oxide on the skin and eyes of 44 patients with PXE. Results, however, were inconclusive and, especially for the ophthalmologic changes, unpromising [163]. 

#### 3.4.3. Tocopherol Acetate and Ascorbic Acid

The idea of using tocopherol acetate (TA, vitamin E) and ascorbic acid (AA, vitamin C) originated from Pasquali-Ronchetti and his associates, who hypothesized that oxidative stress is a possible cause of the PXE phenotype [164]. The authors reported significantly elevated levels of reactive oxygen species in PXE fibroblasts compared to controls. Since then, two case reports have described attempts at treating PXE via supplementation with TA and AA [165,166]. The report by Akoglu et al. describes the influence of TA and AA supplementation on the clinical and histopathological features [166]. The authors observed a clinical and histological reduction of skin changes without further aggravation of the fundoscopic findings. However, after one year of treatment, lesions were starting to progress again. This is in line with the results of Li et al., who reported elevated levels of reactive oxygen species (ROS) in *ABCC6_−/−_* mice. An antioxidant diet (TA, AA, selenium, N-acetylcysteine) was able to reduce ROS levels, but failed to influence connective tissue calcification in these mice [167].

#### 3.4.4. Vitamin K

A possible role for vitamin K in the pathogenesis of PXE has been discussed, at least since Vanakker et al. reported a series of seven patients with severe skin changes (“cutis laxa”), who were initially diagnosed with PXE via skin biopsy, yet showed no vascular or ocular manifestations. Genetic diagnostics revealed a PXE-like genotype in six out of seven patients caused by pathogenic variants of the *GGCX* (gamma-glutamyl carboxylase) gene [168]. The *GGCX* gene encodes the endoplasmic enzyme gamma-glutamyl carboxylase (GGCX) protein, which is necessary for the post-translational carboxylation of glutamate and, thus, activation of various vitamin-K-dependent proteins (VKDPs) [169]. The group of VKDPs encompasses important proteins, such as coagulation factors II, VII, IX, and X, matrix gla protein (MGP), gla-rich protein, and osteocalcin (OC) [169]. Vitamin K serves as a cofactor in carboxylation, and has to be “recycled” via VKORC1 afterwards [170]. As mutations in both the *GGCX* and *VKORC1* genes have been associated with PXE-like changes or soft connective-tissue calcification [148,171], respectively, vitamin K was assumed early on to play a role in PXE pathology [140,148]. This theory was substantiated by findings indicating a reduced activity of VKDPs in patients with PXE [172], and a reduction of ectopic mineralization in a zebrafish model of PXE via administration of vitamin K [173]. However, despite the circumstantial evidence of vitamin K deficiency causing PXE, vitamin K is neither a substrate of the ABCC6 transporter [174], nor does vitamin K supplementation counteract ectopic calcification in *ABCC6_−/−_* mice [175,176,177] or PXE humans [178].

Nonetheless, the pro-calcifying potential of vitamin deficiency and VKDP suppression should not be underestimated in PXE: pathogenic variants of the *GGCX* or *VKORC1* genes may play a role in patients with severe or early onset disease; the usage of vitamin K antagonists should only be considered with extreme caution.

#### 3.4.5. Conclusions

Administration of phosphate binders, Mg^2+^, or vitamins do not result in a significant reduction of disease activity in patients with PXE. Although mouse models showed promising preliminary results, the administration of these agents to PXE patients are not a real therapeutic option, far less a cure. Effects, if any, were temporary and limited to the skin. Nonetheless, all of these agents are connected with tissue or vascular calcification. Therefore, a supportive protective role of Mg^2+^ or a phosphate binder should be considered in the treatment of PXE patients.

### 3.5. Pyrophosphate, Bisphosphonates, and the Direct Inhibition of Hydroxyapatite Formation

#### 3.5.1. Pyrophosphate and the Inhibition of Tissue Non-Specific Alkaline Phosphatase (TNAP)

The role of pyrophosphate (PPi) as a major inhibitor of ectopic calcification was first suggested by Fleisch et al. in 1962 [179], and has been widely acknowledged since then [180]. Physiologically, mineralization of soft connective tissue and vessels is inhibited by numerous factors, of which pyrophosphate is one of the most potent. To induce calcification in hard tissues such as bone and teeth, PPi is actively hydrolyzed by local phosphatases such as TNAP [181]. Linking PPi to inherited diseases that are characterized by progressive calcification was first suggested by Stuart in 1993, who found reduced levels of PPi in an infant with GACI [182]. PPi deficiency was first discussed as the major cause of ectopic calcification in patients with PXE [183] by Jansen et al. in 2013 [184]. They found ABCC6-dependent regulation of PPi in *ABCC_−/−_* mice. Since then, research has focused on re-establishing PPi homeostasis, either by supplementing patients with pyrophosphate(-analogs) or by the inhibition of PPi hydrolyzation (Table 3). 

Despite its potential benefits, treatment of vascular calcification by the administration of PPi was considered impractical for a long time [184,185]. In particular, the short half-life (30 min) of PPi, and the assumption that it would be rapidly hydrolyzed after oral administration made PPi unattractive as a therapeutic agent [185]. Therefore, early experiments administered PPi to rats via subcutaneous or intraperitoneal injection [186]. For example, Pomozi et al. administered daily injections of pyrophosphate to *ABCC6_−/−_* mice, resulting in a near complete suspension of ectopic calcification, despite a poor PPi bioavailability of only about 0.5% [187]. The assumption of the negligible bioavailability of orally administered PPi was proven wrong by Dedinszki et al. in the same year [188]. They provided *ABCC6^−/−^* mice with oral PPi via drinking water and also found a near complete attenuation of soft connective-tissue calcification. In addition, the authors were able to show an increase in the serum concentration of orally administered PPi in humans [188]. This result was successfully reproduced in 2019 when Pomozi et al. discovered a doubling of the concentration of serum PPi and a reduction of ectopic calcification in their *ABCC6_−/−_* mice after feeding them a new PPi-enriched diet [189]. The authors further concluded that, despite a low bioavailability of PPi after oral administration (around 0.1%), the differences in individual dietary habits of PXE patients may significantly contribute to their phenotypic heterogeneity [189]. 

Another possibility to raise peripheral PPi levels is to inhibit its hydrolyzation. Physiologically, mineralization occurs in two steps: first, hydroxyapatite crystals form within the matrix vesicles; in a second step, hydroxyapatite is propagated through the vesicle membrane into the extracellular matrix. To accumulate and further elongate in the extracellular matrix, hydroxyapatite crystals depend on high concentrations of calcium and phosphate. The ratio of phosphate to inorganic pyrophosphate is especially crucial (Pi/PPi-ratio) [190]. Therefore, local hydrolyzation of PPi is a key-step in the progressive mineralization of tissues. Hydrolyzation of PPi is catalyzed by alkaline phosphatases, of which tissue non-specific alkaline phosphatase (TNAP) is the most important [181,190,191]. Upregulation of TNAP has been connected to vascular calcification in renal failure [191]. Furthermore, pathogenic variants of *NT5E* encoding CD73, an enzyme that hydrolyzes adenosine monophosphate (AMP) to adenosine and phosphate, led to increased TNAP activity, resulting in a phenotype similar to PXE, with severe calcification of lower limb arteries [192]. Ziegler et al. (2017) provided evidence of increased expression and activity of TNAP in PXE fibroblasts [193]. Furthermore, the authors were able to prevent calcification in vitro via inhibition of TNAP, by administering SBI-425. SBI-425 is an arylsulfonamide derivate with optimized pharmacogenetic properties [193,194]. Next, *ABCC6_−/−_* mice were fed with SBI-425 (30 mg/kg body weight and day) and control *ABCC6_−/−_* mice were fed with etidronate (240 mg/kg body weight and day), as well as *ABCC6_−/−_* mice on a control diet. SBI-425 administration resulted in attenuation of both the development and progression of ectopic calcification that was comparable to etidronate treatment, yet without an influence on bone microarchitecture. In 2019, Li et al. verified these results [195]. 

In conclusion, orally administered PPi may be a promising approach to treat ectopic calcification in patients with PXE. Especially since PPi is a common food additive that is considered nontoxic by the US Food and Drug Administration and the European Union, it would be a cost-efficient and simple treatment of a, so far, incurable disease. However, with age or comorbid renal disease, treatment of PXE patients with PPi may become a problem, as it is hydrolyzed to monophosphate, which is considered to be one of the main determinants of vascular calcification in patients with CKD [196]. A combination therapy with TNAP inhibitors may be a conceivable approach in this context. However, a trial with administration of TNAP inhibitors to PXE patients has yet to be initiated.

#### 3.5.2. Bisphosphonates

Bisphosphonates (BP) are stable, non-hydrolyzable analogs of PPi. They are characterized by two phosphate groups, covalently bonded to a central carbon. Two side chains further determine the chemical characteristics of the different BPs. BPs are classified into two categories according to the structure of their side chains and their molecular mode of action: simple or non-nitrogen-containing BPs (S-BPs) and nitrogen-containing BPs (N-BPs). The first group encompasses the earliest generation of BPs, while the second group contains newer generations of BPs, with a higher affinity for bone tissue, and an up to 10,000-fold higher effectiveness for the prevention and therapy of osteoporosis, as compared to etidronate [197,198]. The first report of a possible inhibitory effect of BPs on vascular calcification was published in 1970 [199]. Since then, various studies have focused on the effect of BPs on vascular calcification [197,200] (Table 3).

The usage of BPs in the therapy for PXE has been discussed since Jansen et al. reported reduced serum levels of PPi due to ABCC6 deficiency [184]. In their publication, the authors proposed treatment with BPs, since these agents had already been successfully used in patients with GACI [201,202]. In GACI, infantile arterial calcification is also associated with reduced levels of PPi due to a pathogenic variant of *ENPP1* [203]. Animal studies have suggested an inhibitory effect on vascular calcification for both S-BPs and N-BPs. However, etidronate, the first and simplest S-BP, was the most potent inhibitor of vascular calcification [197,200]. This effect was mirrored in animal studies on *ABCC6_−/−_* mice. Li et al. (2015) found etidronate, but not alendronate (N-BP), significantly reduced ectopic mineralization in *ABCC6_−/−_* mice [204]. However, the authors used a 12-fold higher dosage of BPs than the dosage commonly used in humans. In 2018, the authors published a second similar study, but using reduced dosages of etidronate (0.12-fold) [205]. Here, etidronate also prevented, but did not reverse, ectopic mineralization. Similar effects occurred in *ABCC6_−/−_* mice with induced dystrophic cardiac calcification (DCC); etidronate treatment nearly abolished DCC, whereas administration of alendronate had no significant effect [187]. However, N-BPs do not necessarily have no effect on ectopic calcification. At least in PXE fibroblasts the N-BP zoledronate was able to attenuate calcification [206].

The TEMP-Trial (Treatment of Ectopic Mineralization in Pseudoxanthoma elasticum trial), published in 2018 by Kranenburg and his colleagues, was the first and, up to now, the only randomized, double-blinded, placebo-controlled, single-center study administering etidronate to patients with PXE. Participating patients were cyclically administered either etidronate (20 mg/kg) or a placebo for two weeks every 12 weeks for one year [207]. The primary outcome was change a in femoral ^18^flouride sodium positron emission tomography (^18^F-NaF PET). Secondary outcomes were the computed tomography (CT)-based change in femoral arterial calcification, sonographic assessment of intima-media thickness, and a number of CNV events (e.g., intravitreal VEGF injection). The authors observed no significant difference regarding femoral artery calcification, assessed via ^18^F-NaF PET; however, the CT-based quantification of femoral artery calcification was significantly lower in patients treated with etidronate [207]. In a post-hoc analysis, the authors specified that the administration of etidronate halted vascular calcification in all vascular territories except the coronary arteries [208]. An initially reported significant benefit of the occurrence of CNV events [207] was countered by Risseeuw et al., who saw no protective or deteriorating effect of etidronate on CNV activity, after an adjustment for the baseline burden of CNV [209].

In summary, despite encouraging data derived from studies on *ABCC6_/−_* mice, results of the first trial of BP treatment that included PXE patients were mixed. Although etidronate seemed to attenuate, or even reduce arterial calcification, no effect on cardiac vessels or CNV was observed. Therefore, the most dangerous and the most quality-of-life-limiting characteristics of PXE were both unaffected by this treatment. However, more research is needed to really evaluate the role of bisphosphonates in the therapy for PXE. The jury is still out on etidronate.

In this context, another clinical trial is currently planned in Angers, France [210]. The authors intend to test orally administered PPi vs. etidronate (https://clinicaltrials.gov/ct2/show/study/NCT04868578; accessed on 1 November 2021).

#### 3.5.3. Phytate

Phytate (myo-inositol hexakisphosphate, IP6) is a ubiquitously occurring inositol phosphate that was initially discovered in plants. In mammals, IP6 is involved in various processes, such as phosphatase inhibition, nuclear transport, and vesicular recycling [211]. As an intravenously administered agent, IP6 inhibits the formation and growth of hydroxyapatite crystals with good safety and tolerability [212,213,214]. However, IP6 only shows modest potency and a short plasma half-life of a few minutes [215]. Therefore, Schantl et al. generated the inositol phosphate analog (OEG2)2-IP4, with more favorable pharmacokinetics and an extended half -life, which is distributed in the US as INS-3001 [214]. INS-3001 was administered subcutaneously to *ABCC6_−/−_* mice, either continuously via subcutaneous implanted osmotic pumps, or via thrice or once weekly subcutaneous injection [216]. At a dosage of 4 mg/kg/day of INS-3001, by either continuous or cyclical administration, ectopic calcification was prevented or preserved.

IP6 and INS-3001 represent a relatively new but promising therapeutic approach. Further investigations are needed, though, to evaluate their possible clinical use in the future.

#### 3.5.4. Sodium Thiosulfate

Sodium thiosulfate (Na_2_S_2_O_3_, NaTS) is an inorganic salt with a long history in medicine. Initially, it was developed as an antidote for cyanide poisoning [217]. NaTS is a chelating agent that forms highly soluble calcium thiosulfate complexes [218]. Its administration in end-stage renal disease patients resulted in a significantly reduced calcification of the iliac artery and cardiac valves [218]. As NaTS has also been effectively applied in the treatment of patients with calciphylaxis [219,220,221], it has recently been administered to a patient with severe and early onset PXE due to PXE/GACI-overlap-syndrome [144]. After six months, calcific stenoses of the coeliac and mesenteric arteries were no longer detectable; clinical symptoms such as pre- and postprandial pain were resolved. However, symptoms relapsed after the treatment was discontinued [144].

At present, the data does not support a general administration of NaTS to patients with PXE. However, in very severe cases or advanced arterial calcification, NaTS administration may complement other therapeutic approaches, especially due to its potential to reduce existing vascular calcification.

#### 3.5.5. Conclusions

Drug therapy to terminate the ectopic formation of hydroxyapatite crystals has been the focus of research for the last decade. Administration of PPi seems to be an especially promising treatment option for patients with PXE. 

### 3.6. Gene Therapy and ABCC6 Targeted Therapy

#### 3.6.1. Directed Gene Transfer

Theoretically, the easiest way to cure a hereditary disease is to replace the mutated gene with a wild type one. A study investigating the possibility of an adenovirus-mediated delivery of a wild type human *ABCC6* to the liver of *ABCC6_−/−_* mice resulted in a sustained high expression of human ABCC6 protein for up to four weeks [222]. PPi levels increased and were even completely restored to normal levels after repeated injections every four weeks. Subsequently, ectopic mineralization was significantly decreased, but existing mineralization persisted [222]. Adenovirus-based vectors that are delivered to the nucleus of hepatocytes, are one of the most efficient gene-delivery systems [223], as they elicit only a limited immune response, and are considered safe, combined with their ability to infect dividing and quiescent cells [223,224]. Although gene therapy has encountered various setbacks along the way, a report by Huang et al. [222] indicates that in the future a cure for monogenetic PXE could be possible.

In this context, recent progress with improved mRNA-based gene-delivery systems may provide another way of administering PXE gene therapy [225].

#### 3.6.2. Sodium 4-Phenylbutyrate

Approximately, 2/3 of all known disease-causing pathogenic variants of *ABCC6* are missense mutations [226]. The proteins that result from translation of these missense mutations often maintain—at least partly—their transport activity [227,228]. However, most of these mutant proteins are mislocated and do not reach the plasma membrane [229,230]. As the localization of transmembrane proteins within the plasma membrane is crucial to their physiological function, re-localization of the mutant proteins could rescue their remaining functionality and restore their potential to inhibit calcification [227].

**Table 3 biomedicines-09-01895-t003:** Systemic therapy and new therapeutic advances.

Treatment/Administered Drug	Rationale	References
Supplemental and dietary approaches		
Reduced calcium intake and administration of phosphate binders	Regulation of calcium and phosphate as the main substrates of hydroxyapatite formation	[149,151,152,153]
Magnesium	Inhibition of hydroxyapatite formation and reduction of osteogenic expression in vascular smooth muscle cells	[151,157,158,159,160,161,162]
Tocopherol acetate and ascorbic acid	Reduction of oxidative stress	[164,165,166,167]
Vitamin K	Mutations in genes encoding vitamin-K-dependent enzymes result in a PXE-like phenotype; reduced levels of vitamin K promote arterial calcification	[140,148,171,172,173,174,175,176,177,178]
Pyrophosphate, bisphosphonates, and direct inhibition of hydroxyapatite formation		
PPi supplementation	Restoration of serum PPi levels	[186,187,188,189]
TNAP inhibition	Suppression of PPi hydrolyzation	[192,193,194,195]
Pyrophosphate administration	Administration of stable PPi analogues; suppression of plaque calcification	[201,202,204,205,206,207,208,209]
INS-3001	Inhibition of hydroxyapatite formation	[216]
Sodium thiosulfate	Calcium chelating and reduction of calcification	[144,219,220,221]
Gene therapy and ABCC6 target therapy		
Directed gene transfer	Adenovirus-mediated delivery of wild-type *ABCC6* to the liver	[222]
Sodium 4-phenylbutyrate	Re-localization of misfolded ABCC6 proteins to the plasma membrane	[227,228]
PTC-124	Readthrough of premature stop codons	[231]

Abbreviations: PXE, Pseudoxanthoma elasticum; PPi, pyrophosphate; TNAP, tissue non-specific alkaline phosphatase.

Sodium 4-phenylbutyrate (4-PBA) is an aromatic, fatty acid used in the treatment of urea cycle disorders [229]. 4-PBA also acts as a chemical chaperone and corrects trafficking defects of misfolded proteins, including mutated proteins associated with human diseases [227]. In this context, a possible beneficial effect of 4-PBA in *ABCC6_−/−_* mice expressing two different mislocalized human ABCC6 mutants, was first described in 2011 [228]. The experiment was expanded in 2014 through the utilization of a zebrafish model of PXE. Herein, an additional four ABCC6 mutants, with preserved transport ability, were restored to the plasma membrane by the administration of 4-PBA [228]. Additionally, there is evidence of a restored inhibition of calcification in *ABCC6_−/−_* mice expressing human ABCC6 mutants and treated with 4-PBA [227].

However, although an intriguing concept, implemented in the therapy of many hereditary diseases, such as cystic fibrosis and Fabry’s disease, the administration of 4-PBA can only be considered in PXE patients with mislocated ABCC6 mutants that have (partly) preserved transport activity. 

#### 3.6.3. PTC-124

Nonsense mutations account for approximately 35% of pathogenic *ABCC6* variants, of which p.Arg1141* is the most common in the Caucasian population [232]. Nonsense mutations occur when an amino acid encoding codon is transformed to a stop codon, resulting in a prematurely terminated translation and, subsequently, a truncated protein, lacking its physiological function [233]. It has been observed that the recoding of such a premature stop codon (PTC) occurs in about 1% of translation events, whereas suppression of a normal stop codon is much less frequent [233]. This suggested a novel therapeutic approach, which attempts to increase the chances of PTC suppression. One of the most promising compounds for this approach is 1,2,4-oxadiazole (PTC-124), which was first described in 2007 [234]. Although PTC-124 has been evaluated in phase II and III clinical studies with mixed results [233,235,236,237], is has been approved by the European Medicines Agency for the treatment of Duchenne muscular dystrophy, which is also caused by a premature stop codon. In this context, PTC-124 was tested via in vitro incubation, with HEC293 cells transfected with seven different *ABCC6* nonsense variants, including p.Arg1141* [231]. Hereby, the amount of functional, full-length proteins was significantly increased. However, further studies utilizing *ABCC6_−/−_* mice with relevant clinical endpoints, such as calcification or serum concentration of pyrophosphate, are still lacking.

Therefore, PTC-124 bears some potential for PXE patients who have at least one nonsense mutation. However, further studies are needed and, given the mixed results of phase III clinical studies, expectations should not be too high. 

#### 3.6.4. Conclusions

In the future, direct gene transfer may hold the best potential of all the therapeutic approaches for PXE. Although the concepts are intriguing, 4-PBA or PTC-124 are unlikely to be administered to patients with PXE, as other treatment options are more likely to bridge the time until direct gene transfer is feasible.

## 4. Final Conclusions

Current therapeutic options target the main hallmarks of PXE. Skin lesions should be treated conservatively. At present, there is not sufficient evidence to support cosmetic surgery as a regular treatment option in PXE. Therefore, surgery should only be considered with caution and after carefully weighing of the operative risks and individual suffering. Due to the effectiveness and safety of intravitreal administrated anti-VEGF agents to treat choroidal neovascularization in PXE, it has rapidly become the state-of-the-art treatment, since its introduction in 2005. Anti-VEGF treatment should be started as early as possible in PXE patients, and regular examinations by ophthalmologists specialized in PXE are essential. Treatment of vascular calcification is currently based on three pillars: medical treatment, walking exercise, and interventional therapy. Medical treatment mainly consists of lipid-lowering drugs, favoring atorvastatin. Walking exercise, especially SET, promotes muscle capillarization and induces collateralization. Therefore, it significantly reduces intermittent claudication and enhances quality-of-life. Although regularly performed in clinical routine, reports on interventional therapy for both PAD and CAD are rare. Following the current guidelines, revascularization should be performed, if necessary, and strictly accompanied by walking exercise and medical treatment.

Future therapeutic possibilities mainly comprise two different approaches. Gene therapy and directed gene transfer, in particular, hold the most potential to cure PXE in the long-term. However, various influences of genetic interaction or modifiers still need to be illuminated before this becomes a reality [238,239,240]. Reconstitution of PPi homeostasis, by either PPi supplementation or pyrophosphate administration, will likely become part of medium-term PXE treatment. Although research is just getting started in this area, the results are, so far, promising. 

Given the complex pathogenesis of PXE, including disturbed PPi homeostasis and dyslipidemia, combination therapies seem to be most promising. Therefore, future studies should focus on multimodal types of treatment. For example, combination therapy of atorvastatin and etidronate has already proven effective in patients with atherosclerosis [241].

As a final sentence, all of the surveyed literature can be summarized in one question: “Is there the one treatment for PXE?” The answer is probably “no”. 

## Figures and Tables

**Table 1 biomedicines-09-01895-t001:** Intravitreal injection of vascular endothelial growth factor inhibitors.

Author (Year)	Design	Population	PXE (%) *	Follow-Up [Months] (Mean)	Treatment	Summary
Teixeira et al. (2006) [68]	Case report	1 patient (1 eye)	0	3.5	Bevacizumab (2 × 1.25 mg)	First report of intravitreal injection of anti-VEGF for treatment of CNV secondary to angioid streaks Improvement of VA
Lommatzsch et al. (2007) [69]	Case report	1 patient (1 eye)	1 (100)	N/A	Bevacizumab (3 × 1.25 mg)	Preceding multiple unsuccessful treatments with V-PDT combined with intravitreal injection of triamcinolone. Improved VA and reduced disease activity after anti-VEGF treatment
Bhatnagar et al. (2007) [70]	Retrospective case series	9 patients (9 eyes)	9 (100)	4–8 (6)	Bevacizumab (mean: 1.8 × 1.25 mg)	Improvement or stabilization of VA in all treated eyes No treatment-related adverse events
Rinaldi et al. (2007) [71]	Retrospective case series	5 patients (5 eyes)	5 (100)	3–9 (6)	Bevacizumab (1–2 × 1.25 mg)	Initial improvement of VA and stabilization thereafter
Finger et al. (2007) [72]	Prospective case study	15 patients (16 eyes)	15 (100)	2–20 (8)	Bevacizumab (monthly, PRN, mean: 2.4 × 1.5 mg)	Initial improvement of VA and stabilization thereafter The fewer fundoscopic changes at baseline, the more efficient is anti-VEGF treatment No treatment-related adverse events
Schiano et al. (2008) [73]	Case report	2 patients (2 eyes)	2 (100)	14	Bevacizumab (3 × 1.25 mg)	Recurrent decrease of VA after initial injection, continuing improvement after re-injections
Wiegand et al. (2008) [74]	Retrospective case study	6 patients (9 eyes)	2 (33)	10–28 (19)	Bevacizumab (mean: 4.4 × 1.25 mg)	Improvement or stabilization of VA in 8 of 9 eyes No treatment-related adverse events
Neri et al. (2009) [75]	Prospective case study	11 patients (11 eyes)	N/A	20–29 (24)	Bevacizumab (monthly, PRN, mean: 3.5 × 1.25 mg)	Stable or improved VA in all treated patients for up to 29 months No treatment-related adverse events
Sawa et al. (2009) [76]	Retrospective case series	13 patients (15 eyes)	7 (54)	12–24 (19)	Bevacizumab (mean: 4.5 × 1.0 mg)	Stabilization of VA CNV recurrence in 5/15 eyes, development of new CNV in 3/15 eyes Cataract surgery in one eye due to lens injury during the injection
Myung et al. (2010) [77]	Retrospective case series	9 patients (9 eyes)	9 (100)	24–31 (29)	Bevacizumab (1.25 mg) or ranibizumab (0.5 mg)	Initially, all patients received bevacizumab, 4 patients switched to ranibizumab 4 patients with maintenance regimen (1 injection every two months) and 5 patients with PRN regimen Improvement or stable VA in patients with treatment-naïve eyes or good VA at baseline rather than in patients with poor VA at baseline
Ladas et al. (2010) [78]	Prospective case study	14 patients (15 eyes)	3 (21)	12	Ranibizumab (7 × 0.5 mg, “inject and extend”)	Improvement or stabilization in 14/15 eyes No treatment-related adverse events reported
Finger et al. (2011) [79]	Prospective clinical trial	7 patients (7 eyes)	7 (100)	12	Ranibizumab (monthly, 12 × 0.5 mg)	Decrease of central retinal thickness and fluorescein leakage after the first injections Improvement of VA mainly in patients with early disease stages and absence of irreversible damage
Finger et al. (2011) [80]	Retrospective case series	14 patients (16 eyes)	14 (100)	Mean: 23–32	Bevacizumab (monthly, PRN, 6.5 × 1.5 mg)	Two groups according to fundus changes: early and advanced Better outcome in patients with early disease stages compared to advanced Suspicion of possible RPE-atrophy as a long-term adverse effect
Zebardast and Adelman et al. (2012) [81]	Case report	1 patient (1 eye)	1 (100)	5 years	Ranibizumab (monthly for 12 months, PRN afterwards, 14 × 0.5 mg).	Improvement and stabilization of the affected left eye No treatment-related adverse events during follow-up
Alagöz et al. (2015) [82]	Retrospective case series	20 patients (23 eyes)	10 (50)	6–58 (23)	Bevacizumab (PRN, mean: 5.1 × 1.25 mg)	More aggressive course of CNV in younger individual with more resistance to treatment Insignificantly more injections per month in PXE eyes (*p* = 0.072)
Rosina et al. (2015) [83]	Retrospective case study	10 patients (16 eyes)	N/A	30–67 (52)	Bevacizumab (PRN, 2.5 × 1.25 mg)	Low rate of recurrences in patients free from CNV reactivation for one year Recurrence increased the risk for another recurrence No mean improvement of final VA compared to baseline
Ladas et al. (2016) [84]	Prospective case study	19 patients (20 eyes)	3 (14)	24	Ranibizumab (9 × 0.5mg, “inject and extend”)	2-year results of previously mentioned study After initial improvement of VA during the first year, maintenance of VA during the second year No treatment-related adverse events
Mimoun et al. (2017) [85]	Multicenter, retrospective, observational study	72 patients (98 eyes)	72 (100)	48	Ranibizumab (mean annual: 4.1 × 0.5 mg)	Largest evaluation of anti-VEGF treatment in PXE patients up to now Widely improved or stabilized VA Recurrence of CNV in ~30% per year; re-treatments and follow-up examinations necessary Treatment-related adverse events (amongst others): eye pain (4.2%), VA decrease (2.8%), amaurosis (1.4%), cataract (1.4%), ischemic cerebral infarction (1.4%), infections (2.8%)
Lekha et al. (2017) [86]	Retrospective case series	10 patients (15 eyes)	3 (30)	25–100 (57)	Bevacizumab (PRN, mean: 5.6 × 1.25 mg)	Improvement or stabilization in 73%; any recurrence in 73%
						Correlation between number of injections and recurrence rate
Gliem et al. (2019) [87]	Prospective clinical trial	15 patients (15 eyes)	15 (100)	12	Aflibercept (initial injection of 2 mg, PRN thereafter, mean: 6.7 × 2 mg)	Improvement of VA mainly in patients with low initial VA Adverse events: increased ocular pressure (2 patients), conjunctival hemorrhage (2 patients), foreign body sensation (1 patient)
Sekfali et al. (2020) [88]	Retrospective case series	13 patients (14 eyes)	N/A	12	Switching from ranibizumab to afliberzept (dose not reported; most likely 6.6 × 2 mg)	Included patients sustained recurrent or persistent CNV while treated with ranibizumab After switching: VA stabilized in 10 eyes, improved in 2 eyes, and declined in 2 eyes Adverse events: one patient suffered massive intraretinal hemorrhage after 11 months Switching of applicated agent may be a therapeutic option in recurrent or persistent CNV

* number of patients with PXE included in the particular report. Abbreviations: PXE, Pseudoxanthoma elasticum; CNV, choroidal neovascularization; VA, visual acuity; V-PDT, verteporfin photodynamic therapy; VEGF, vascular endothelial growth factor; PRN, pro re nata; N/A, not available.

**Table 2 biomedicines-09-01895-t002:** Interventional and surgical treatment of peripheral and coronary artery disease.

Author (Year)	Design	Population	Treatment	Summary
Interventional/surgical treatment of peripheral artery disease				
Carter et al. (1976) [141]	Case report	1 patient	Femoropopliteal bypass grafting	Distal occlusion of left femoral artery and successful femoropopliteal bypass grafting with autogenous saphenous vein Patent bypass after 12 months
Slade et al. (1990) [106]	Case report	1 patient	PTA	High grade stenosis of right femoral artery and successful recanalization via PTA No long-term follow up
Rühlmann et al. (1998) [142]	Case report	1 patient	Surgical endarterectomy	Multi-segment arterial occlusion of both femoral arteries (right side with patent femoropopliteal bypass) Surgical endarterectomy with satisfying primary result Re-occlusion after 4 months and conservative treatment thereafter
Ammi et al. (2015) [143]	Retrospective case series	4 patients	PTA with and without stenting; femoropopliteal bypass grafting; conservative proceedings	Four cases with initial PTA with and without stenting followed by re-stenosis/-occlusion. Due to high failure rate no recommendation of PTA by the authors
Interventional/surgical treatment of coronary artery disease				
Bete et al. (1975) [105]	Case report	1 woman (18 years old)	CABG	Severe triple-vessel CAD with pronounced collateralization Successful CABG with two autologous vein grafts Symptom-free at 12-month follow-up
Slade et al. (1990) [106]	Case report	1 man (31 years old)	Conservative proceeding	Medial occlusion of LAD and collateralization via right coronary artery Settling of symptoms after medicinal treatment
Alexopoulos et al. (1994) [111]	Case report	1 woman (29 years old)	Conservative proceeding	Severe triple-vessel CAD with multiple collateral vessels without clinical or angiographic signs of infarction with concomitant elevation of lipoprotein(a)
Kévorkian et al. (1997) [107]	Case report	1 woman (18 years old)	Conservative proceeding	Severe triple-vessel CAD Conservative, medicinal treatment due to lack of interventional or surgical possibilities Symptom-free after 2 years
Sarraj et al. (1999) [113]	Case report	1 man (61 years old)	CABG	Severe triple-vessel CAD with occlusion of LAD and RCA and high-grade stenosis of CX Successful coronary bypass surgery; LIMA during operation with reduced flow due to atherosclerotic stenosis Histological sampling of LIMA with atherosclerotic lesion consistent with PXE
Araki et al. (2001) [108]	Case report	1 man (63 years old)	PCI	History of early myocardial infarction and stroke at the age of 38 and consecutive double coronary artery bypass grafting Severe triple-vessel CAD and stenosis, respectively occlusion, of coronary bypass grafts Successful PTCA with stenting of right and left coronary arteries and one bypass.
Song et al. (2004) [116]	Case report	1 woman (67 years old)	CABG	Severe triple-vessel disease with history of myocardial infarction and multiple peripheral vascular procedures Successful CABG using LIMA and both radial arteries as no sufficient saphenous vein conduit existed After one year, development of chest pain; LIMA bypass was patent, but occlusion and high-grade stenosis of radial artery grafts; symptom relief after PTCA and stenting radial arterial grafts
Baglini et al. (2005) [110]	Case report	1 man (52 years old)	PCI	Medial stenosis of LAD branch; treatment via implantation of a heparin-coated stent 12-month follow-up without any signs of re-occlusion
Kocaman et al. (2007) [109]	Case report	1 woman (21 years old)	CABG	Severe triple-vessel CAD with anomalous origin of the RCA from the left coronary sinus Successful coronary bypass surgery with correction of the right coronary artery Uneventful postoperative course
Rossiter-Thornten et al. (2011) [112]	Case report	1 man (56 years old)	CABG	Severe triple-vessel CAD; no signs of atherosclerosis in LIMA Successful CABG; histological sampling of LIMA with patchy, non-narrowing, calcific changes Evaluation of arterial grafts prior to surgery recommended by the authors
Sasai et al. (2012) [114]	Case report	1 woman (26 years old)	PCI	Significant stenosis of medial LAD and CX without visualization of coronary calcification in computed tomography Despite peri-interventional dilation and post-dilations after deployment of a bare metal stent with noncompliant balloons, residual under-expansion of the stent
Karam et al. (2015) [117]	Case report	1 woman (42 years old)	PCI	Severe stenosis of LAD and nonsignificant stenosis of RCA PCI with implantation of one stent; sequential, high-pressure (30 atmospheres) inflations with a noncompliant balloon were necessary
Anzai et al. (2017) [115]	Case report	1 man (58 years old)	PCI, POBA	High grade stenosis of RCA and chronic total occlusion of LAD Pre-dilation of RCA and implantation of two everolimus-coated stents with no residual flow limitation; recanalization of LAD and performing of POBA with resulting TIMI III flow.

Abbreviations: PXE, Pseudoxanthoma elasticum; PTA, percutaneous transluminal angioplasty; CABG, coronary artery bypass grafting; PCI, percutaneous coronary intervention; CAD, coronary artery disease; LAD, left anterior descending coronary artery; RCA, right coronary artery; CX, left circumflex coronary artery; LIMA, left internal mammary artery; POBA, plain old balloon angioplasty; TIMI, thrombolysis in myocardial infarction.

## Data Availability

Not applicable.

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
