# Peer review of "Therapy of Pseudoxanthoma Elasticum: Current Knowledge and Future Perspectives"

_biomedicines, 2021, doi:10.3390/biomedicines9121895_

Round 1

Reviewer 1 Report

The manuscript written by Dr Max Jonathan Stumpf and colleagues is a much awaited and excellent review of currently available treatments and unmet medical needs for patients affected with Pseudoxanthoma elasticum.

The paper is well structured and a clear and concise conclusion ends every chapter.

The reference section is very good.

In order to further improve the quality of the manuscript I have both major and minor remarks:

MAJOR

Editing of English language is required.

MINOR

1° Abstract and all along the texte : use ‘pathogenic variant’ instead of ‘mutation’

2° Line 14: correct the typo

English language needs to be edited in the abstract

3° line 29: is encoded

4° line 92: I would recommend to add ‘about which the patient should be informed’ or ‘that should not be underestimated’

5° line 94: replace ‘mental’ by ‘chin’

6° line 230: the fact that most reported cases are not necessarily proven to be due to PXE (absence of definitive diagnosis) should be more emphasized in the conclusion of this chapter

7° line 333: use lower case (abcc6 -/-) not upper case to describe the mouse gene

8° line 378: an exemple that English should be improved throughout the manuscript

9° line 471: table 1 and table 2 should be better paginated. It is currently very confusing for the readers

10° line 770: Please write human gene ABCC6 using italics

11° lines 769-771: With respect to 4-PBA or other chaperone therapies, this has been successfully developed and implemented in the clinics for a number of other genetic diseases including cystic fibrosis (Elexacaftor-tezacaftor-ivacaftor) and Fabry disease (Migalastat). In the later, a significant percentage of pathogenic variants are actually 'amenable' to chaperone therapy. Please delete the word 'minority' and rephrase line 770 quoting the aforementioned genetic diseases as successful clinical development programs of chaperon or potentiators.

Author Response

Dear reviewer,

thank you for your encouraging report and your suggestions. I hope, I have adressed them satisfactorily:

Major:

The manuscript has now been proofread by a native speaker (Meghan Lucas), who belongs to our department. Language, style, and typos have been substancially corrected.

Minor:

  1. "Muation" was replaced by "pathogenic variant" where appropriate.
  2. Referring to the points 2, 3, and 8: typos, language, and style were corrected.
  3. Line 92: "about which the patient should be informed" was inserted.
  4. Line 94: "mental" replaced by "chin"
  5. Line 230: The passage has been changes according to your suggestion
  6. Referring to the points 7 and 10: ABCC6, when referrring to the gene, is now always written in italics. ABCC6-/- mice also has been adjusted.
  7. I was not able to improve the pagination of the tables 1 and 2 in the document, but I will suggest it to the editors.
  8. Line 770 has been adjusted according to your suggestions.

I hope, that I was able to adress all your concerns to your full satisfaction. Please let me know, if there is more to improve.

Best regards,

For the authors:

Dr. med. Max Jonathan Stumpf

Reviewer 2 Report

This paper by Stumpf et al, is a good attempt to review and discuss past and current therapeutic solutions for PXE.

This is relatively well written and interesting to a clinical audience, although this reviewer has found numerous typos and errors that will need correcting. Choice of words, sometimes, are a little awkward. A thorough review of references cited should also be done. Many statements are made without proper citations.

Overall, the authors are trying to distinguish their review from two previous ones, clearly cited in the introduction: Germain et al, 2017 and Shimada et al, 2021, by focusing on clinical questions and relevance of therapies. For the most part, the authors have succeeded, however, from Chapter 3.4 on, the text starts to significantly overlap with previous reviews and notably Shimada et al. It would be advisable for the authors to steer clear of what has already been published, and those therapeutic solutions that have currently little to no chance of ever being used such as PCT-124, Phytate, gene therapy (fore the time being), etc… and strictly focus and discuss on these approaches that have clinical prospects: PPi and bisphosphonates. 4-PBA could be mentioned, maybe.

Specific comments:

Lane 39, the very recent Brampton et al, 2021 should also be cited here with respect to atherosclerosis, among others, as it brings a different perspective on the subject. Similarly, Kauffenstein et al 2014 should also be mentioned when it comes to vascular alterations.

The authors mentioned lung dysfunction on lane 41, however, there isn’t any significant pulmonary function changes in PXE and the reference cited is quite clear on that – CO alterations are sub-clinicals.

Chapter 3 (starting lane 69). The authors should clearly distinguish between medial calcification (arteriosclerosis) and intimal calcification. These do not share the same pathophysiology. Furthermore, the author should also distinguish more clearly in their text between pathological calcification (ectopic and dystrophic) and physiological calcification.

In Table 1: what does PXE (%) means? Summary is misspelled.

The conclusion on chapter 3.2.6 is confusing and should be re-written.

Lane 298: no, PXE calcification does not affect all parts of the vascular beds. This is incorrect. Refer to the many papers on PAD in PXE patients.

Throughout the text, referring to PXE mice should either done as Abcc6-/- or ABCC6-/- with proper use of italic. Same goes for citing genes, italic applies.

Author Response

Dear reviewer,

thank you for your contructive and encouraging report and your suggestions. I have adressed your concerns as follows:

Main concerns:

  1. English language, style, and typos: The manuscript has now been proofread by a native speaker; language, style, and typos have largely been corrected.
  2. Citations and references have been checked; citations were amended where they were missing.
  3. Overlap with the previous review by Shimada et al.: I share your concern as there has been little to add since the impressive review by Shimada and collegues. However, besides focusing clinical relevant therapeutic strategies, which has been our main intention, this review intended to comprehensively overview therapeutic strategies for PXE including PCT-124, phytate, gene therapy, etc. as they frequently are requested by patients. This aspect was unclearly stated in the introduction, therefore it has been adjested, accordingly.

Specific comments:

  1. Line 39: Brampton et al. 2021) and Lefteriotis/Kauffenstein et al. (2014) now are mentioned here and when it comes to vascular alterations.
  2. Line 41: Yes, this was unclearly descibed as there is no general impairment of lung functioning in patients with PXE. The statement was clarified.
  3. We tried to distinguish more clearly between intimal and medial calcification (see 3.3.). However, as both entities often are observed in the same vessel, they are both part of the PXE phenotype and can hardly be considered separately.
  4. PXE(%) no is explained below table 1.
  5. After correction of typos, language, and style as well as re-structuring, section 3.2.6 is now more precise.
  6. Lane 298: Lane 298 and similar passages have been corrected.
  7.  ABCC6 is now always written in italics when referring to the gene; PXE mice are now done as ABCC6-/-.

I hope, I was able to adress all concerns to your full satisfaction. Please let me know, if there is more to improve.

Best regards,

For the authors:

Dr. med. Max Jonathan Stumpf